# Evaluation of Transition Metal Catalysts in Electrochemically Induced Aromatic Phosphonation

**DOI:** 10.3390/molecules24091823

**Published:** 2019-05-11

**Authors:** Sofia Strekalova, Mikhail Khrizanforov, Yulia Budnikova

**Affiliations:** Arbuzov Institute of Organic and Physical Chemistry, FRC Kazan Scientific Center of RAS, 420088 Kazan, Russia; so4nar36@yahoo.com (S.S.); khrizanforov@gmail.com (M.K.)

**Keywords:** electrocatalysis, phosphonation, rate constants, cyclic voltammetry, C–H functionalization, metal complex

## Abstract

Voltammetry provides important information on the redox properties of catalysts (transition metal complexes of Ni, Co, Mn, etc.) and their activity in electrocatalytic reactions of aromatic C–H phosphonation in the presence of a phosphorus precursor, for example, dialkyl-*H*-phosphonate. Based on catalytic current growth of oxidation or reduction of the metal catalysts (Co^II^, Mn^II^, Ni^II^, Mn^II^/Ni^II^, Mn^II^/Co^II^, and Co^II^/Ni^II^), quantitative characteristics of the regeneration of catalysts were determined, for example, for Mn^II^, Ni^II^ and Mn^II^/Ni^II^, Co^II^/Ni^II^ pairs. Calculations confirmed the previously made synthetic observations on the synergistic effect of certain metal ions in binary catalytic systems (Mn^II^bpy/Ni^II^bpy and Ni^II^bpy/Co^II^bpy); for mixtures, the observed rate constants, or TOF, were 690 s^−1^ and 721 s^−1^, respectively, and product yields were higher for monometallic catalytic systems (up to 71% for bimetallic catalytic systems and ~30% for monometallic catalytic systems). In some cases, the appearance of pre-waves after adding *H*-phosphonates confirmed the preceding chemical reaction. It also confirmed the formation of metal phosphonates in the time scale of voltammetry, oxidizing or reducing at lower potentials than the original (RO)_2_P(O)H and metal complex, which could be used for fast diagnostics of metal ion and dialkyl-*H*-phosphonate interactions. Electrochemical transfer of an electron to (from) metal phosphonate generates a phosphonyl radical, which can then react with different arenes to give the products of aromatic C–H phosphonation.

## 1. Introduction

Since phosphorus-containing motives are present in a variety of medicinally pertinent synthetic targets, ligand of metal complex catalysts, materials, and so on, development of selective, sustainable, and efficient technologies to construct these kinds of organic structures is of crucial importance [1,2,3,4,5,6,7]. Organophosphorus compounds with P–C bonds containing a phosphonate group P(O)(OAlk)_2_ or phosphonic acid R–PO(OH)_2_ residues are an important class of substrates because they have broad applications in various areas of human livelihood [8,9,10,11,12].

In recent years, researchers have been increasingly interested in direct phosphonation of aromatic compounds (benzene derivatives, *N*,*N*-dialkylbenzamides and nitrobenzene, substituted furans, thiophenes, thiazoles and pyrroles, pyridines and derivatives, etc.) [2,3,13,14,15] by C–H/P–H cross-coupling with dialkyl-*H*-phosphonates and diaryl phosphine oxides [16] during radical initiation of the process or under catalysis of the metal complex (Scheme 1).

The importance of direct C–H aromatic phosphorylation reactions from the point of view of both “green” chemistry and the search for new, more efficient ways of obtaining practically important organophosphorus compounds led to a sharp increase in the number of studies in this area. However, as it turned out, product yields were highly dependent on the reaction conditions, and the optimal oxidizing agents and catalysts were chosen empirically in each case, which was laborious and required screening for each specific substrate [6,13,14,15]. Reaction mechanisms were studied in detail only in rare cases and, as a rule, were postulated. Moreover, the relative catalytic activity of the metal complexes (or salts) in these reactions has not been studied; a rare case assessment has been described for the acetylenes phosphonation reaction, resulting in phosphoryl-containing alkenes [17,18,19]. The relative catalytic activity of metal complexes in these reactions depends on the nature of the latter and reduces in the row Ni > Pd > Rh > Pt. Mechanisms of alkyne phosphorylation catalyzed by transition metal complexes have been proposed [20].

“Metal radical catalysis” [21,22,23,24] or “catalysis in single electron steps” [25] is a concept that combines the advantages of radical chemistry with the privilege of transition metal catalysis and photochemically or electrochemically induced reactions. Implementation of this concept critically depends on the generation of radicals and their conversion to the closed-shell products via reductive elimination or oxidative addition in single electron steps [26]. For both stages, the metal catalyst is, in principle, capable of controlling typical selectivity in the same way as in traditional transition metal catalysis. However, applicability of these catalytic reactions is limited by the redox properties of the catalyst. Therefore, it is highly desirable to expand the “redox potential window” of such catalytic reactions in order to regulate a wide range of processes.

The kinetic patterns of C(sp^2^)–P bond formation reactions by conventional methods have not yet been studied for various reasons, but it is probably because the reactions are slow, carried out at elevated temperatures, and require complex multicomponent mixtures where is difficult to assess the role of each component and its effect on speed. Electrochemical methods are advantageous in the field of synthesis, including mild conditions and exclusion of specially added oxidizing agents and reducing agents [2,27], and can calculate the constants of individual stages of catalytic reactions because a mathematical apparatus is present [28,29,30]. In order to determine the apparent rate constant of a particular reaction and compare the values among them for different catalysts, the reaction scheme, as a rule, is simplified (if its stages are established) and postulated. A highly attractive feature of single-electron step catalysis is that cyclic voltammetry (CV) is suited for prescreening new compounds and efficient reaction conditions [31,32]. CV can be used to characterize the redox-active compounds in solution as well as to study the properties of these compounds and the kinetics of the reactions. Data are being obtained within minutes, and minor amounts of material (mmol) are required, which are essential for catalyst design.

Earlier, we proposed an electrocatalytic approach for the phosphonation of aromatic, heteroaromatic, and heterocyclic compounds in one stage using transition metal complexes and salts as catalysts (Scheme 2) [33,34,35,36,37,38,39,40,41,42,43]. Successful C–H/P–H cross-coupling of dialkyl-*H*-phosphonate was shown with different (hetero)aromatic molecules such as benzenes bearing electron donor and electron withdrawing substituents in the aromatic ring; coumarines under electro-oxidation (Scheme 2A) or electroreduction conditions (Scheme 2B) under the action of Ni, Co, or Mn catalysts and the mixtures of metal complexes [33,34,35,36,37,38,39,40,41,42,43]; and azole derivatives (benzo-1,3-azoles, 3-methylindole, 4-methyl-2-acetylthiazole) under the action of Ag^+^ (Scheme 2C) [20]. It is worth mentioning that electro-oxidative coupling of diphenylphosphine oxide with acetylenes in the presence of catalytic amounts of Ag^+^ yielded the formation of benzo[*b*]phosphole oxides [44].

A radical mechanism type was proposed for the aforementioned electrocatalytic reactions, but the regularities and kinetics have not been established. The purpose of this study is to establish the kinetic patterns and compare the activities of a number of transition metal complexes (Ni, Co, and Mn) or their mixtures in phosphonation reactions of aromatic C–H bonds. In particular, phosphorus-centered radicals are generated using cyclic voltammetry in combination with EPR (electron paramagnetic resonance) spectroscopy to establish the details of the C–H phosphorylation mechanism.

## 2. Results and Discussion

In the studied electrochemically induced reactions, several participants interacted: metal complex-catalyst I, dialkyl-*H*-phosphonate II, and arene III according to the general Scheme 3:

The metal complex (catalyst) is regenerated during the catalytic cycle.

In previous studies on electrocatalytic phosphorylation of C(sp2)–H bonds, metal phosphonates with metal-phosphorus bonds were shown to be key intermediates. The chemistry of metal phosphonates has expanded rapidly in recent years because of their versatile structural characteristics and potential applications [45,46,47,48].

However, the question of the interaction of dialkyl-*H*-phosphonates with salts or metal complexes remains open in the literature since there are few established and fully characterized products. It is interesting that inorganic phosphoric acids (and their derivatives) of type H_2_PO_3_(CH_2_)_n_PO_3_H_2_ [45], ferrocenylenbis (*H*-phosphinic) acid Fc[P(O)H(OH)]_2_ [49], and thiophene-2-phosphonic acid Th[P(O)(OH)_2_]_2_ [50] form metal phosphonates with metal-oxygen coordination, for example, –Co–O–P–O–Co– chains. These compounds are often insoluble metal phosphonate frameworks.

Difficulties with the isolation and establishment of the structure and reactivity of metal phosphonates were determined, as noted [45], for several reasons. The first reason was that growth of single phosphonate-containing crystals was generally more challenging, as they often precipitated rapidly as insoluble, less-ordered phases. While this did not exclude interesting properties, it did make structural characterization of metal phosphonates a challenge. The second reason was that the coordination chemistry of phosphonates was less predictable because other ligating ways were more probable. Therefore, it was interesting to study the system [metal complex-(RO)_2_P(O)H], probably in the presence of aromatics, to compare kinetic parameters for different catalysts and to evaluate the characteristics of catalyst regeneration by means of voltammetry.

We decided to estimate the relative catalytic activity of metal complexes in the studied aromatic phosphorylation reactions using cyclic voltammetry. CV is the most commonly used electroanalytical method to study electrocatalysts since it can be used to determine parameters such as the standard potential (E), the half-wave potential (E_1/2_), the scan rate (v) required for calculating the catalytic turnover frequency (TOF), and the observed rate constant (k_obs_) [28,29,51].

Kinetic parameter TOF quantified the catalytic activity, and the overall rate of homogeneous catalysis was described by the observed (or apparent) rate constant (k_obs_). k_obs_ was useful in order to elucidate the reaction mechanism. This parameter was termed TOF_max_ in some works [28]. Basically, these parameters were calculated in hydrogen oxidation and production reactions, for example in the works of DuBois et al. [30]. Another example where TOF and k_obs_ were calculated was the oxidation of water to O_2_ [52,53]. However, analysis of literature showed that TOF and k_obs_ for catalysts used in phosphonation reactions were not studied.

The catalytic activities of the complexes were measured by successively adding increasing amounts of the phosphorylation agent (EtO)_2_P(O)H to the catalyst solution until the region in which the ratio i_cat_/i_p_ of the catalytic current (i_cat_) (or current of the new wave) to the current of the catalyst oxidation (reduction) in the absence of the phosphorylation reagent (i_p_) was permanently reached. In order to understand which catalyst or catalytic system was the most effective in phosphorylation reactions in terms of speed, we determined the kinetic parameters of catalyst regeneration of MnCl_2_bpy, Ni(BF_4_)_2_bpy, and CoCl_2_bpy, as well as the bimetallic systems MnCl_2_bpy/Ni(BF_4_)_2_bpy, MnCl_2_bpy/CoCl_2_Bpy, and Ni(BF_4_)_2_bpy/CoCl_2_bpy.

### 2.1. Oxidative Conditions

Electrochemical properties of the catalyst complexes were studied in CH_3_CN in the absence and in the presence of increasing amounts of diethyl-*H*-phosphonate (DEP). It was well-known that dialkyl-*H*-phosphonates (RO)_2_P(O)H or diaryl phosphine oxides Ph_2_P(O)H) were not oxidized in the available potential range [19,33,38,39], while their sodium salts, for example, (RO)_2_P(O)Na, were oxidized in the average potential range [54]. Similar observations were made for AgP(O)(OEt)_2_ [20] and Ph_2_P(O)Ag [44] silver salts (Figure 1).

The MnCl_2_bpy complex was oxidized quasi-reversibly in one-electron step at a potential of 0.98 V (Appendix A). When DEP was added to the complex solution, a catalytic growth was observed at practically the same potential. So, at a concentration ratio of catalyst:(EtO)_2_P(O)H = 1:196, the ratio i_cat_/i_p_ = 3.6. The calculated TOF was 355 s^−1^ (Table 1).

Oxidation of the nickel complex Ni(BF_4_)_2_bpy occurred at a potential of 1.83 V (Appendix A). As a result of the addition of DEP to the complex solution, and with further additions, current growth was observed. After a 72-excess of (EtO)_2_P(O)H the current stopped growing. When the concentration ratio of catalyst:(EtO)_2_P(O)H = 1:72, the ratio i_cat_/i_p_ = 1.3. The calculated TOF was 160 s^−1^ (Table 1).

In some cases, adding (EtO)_2_P(O)H to the complex solution led to the appearance of a pre-wave or new wave, the oxidation peak of which was at less anodic potentials; apparently, this was due to a change in the coordination sphere of the metal and the formation of metal phosphonates, which oxidized easier than the initial metal complex. Such a pre-wave was observed for the CoCl_2_bpy complex (Appendix A).

The CoCl_2_bpy complex was oxidized at a potential of 1.4 V. Adding DEP resulted in a new peak at CV at a potential of ≈0.6 V and a catalytic growth at a potential of 1.4 V (Appendix A). The appearance of a new peak appeared to indicate the formation of a new cobalt phosphonate compound [42], which was oxidized easier than the original complex. In this case, the TOF calculation formula could not be used since the observed new wave was not catalytic; it was associated with the chemical reaction of the formation of cobalt phosphonate, and the current reached only one electron level in the presence of excess (EtO)_2_P(O)H. Thus, in this case it was impossible to calculate the TOF or the observed rate constant of catalyst regeneration k_obs_ since the picture differed from the classical one, for which empirical formulas were derived. That is, in this case, under temporary conditions of cyclic voltammetry, catalysis and regeneration of the metal complex along the path of metal-centered oxidation (III)/(II), or in some other way, were not observed with a sufficient, calculable rate.

When the bimetallic catalytic systems MnCl_2_bpy/Ni(BF_4_)_2_bpy and MnCl_2_bpy/CoCl_2_bpy were used, the catalytic current increased at the first oxidation potentials of these systems in the presence of (EtO)_2_P(O)H, and it was more significant compared to the individual complexes. The oxidation of MnCl_2_bpy/Ni(BF_4_)_2_bpy and MnCl_2_bpy/CoCl_2_bpy complexes occurred at potentials of 1.23 V and 1.26 V, respectively. Adding excess amounts of DEP to the mixture of complexes caused a slight displacement of the peaks to the cathode region (Appendix A).

That is, approximately at the oxidation potential of Mn(II)/Mn(III), an intense catalytic growth was observed in the presence of (EtO)_2_P(O)H (Appendix A), and its values were higher than in the case of MnCl_2_bpy (Appendix A). TOF reached the value of 690 s^−1^ for the MnCl_2_bpy/Ni(BF_4_)_2_bpy pair. No calculation could be performed for the MnCl_2_bpy/CoCl_2_bpy pair, as in the case of a cobalt catalyst, a new wave was formed at a potential of ~0.6 V, related to the chemical reaction of cobalt phosphonate formation, although the current increased in the case of a mixture of complexes. CV of a pair of Ni(BF_4_)_2_bpy/CoCl_2_bpy complexes in the presence of DEP is presented in Appendix A, where E_p_ = 1.4 V. After adding 24 equivalents of DEP, the current ceased to increase, and the TOF value reached 721 s^−1^ (Table 1).

Thus, in the case of using a mixture of metal complexes in the presence of (EtO)_2_P(O)H (1:180), i_cat_/i_p_ = 6.1 for the pair Mn^II^(bpy)/Ni^II^(bpy). In comparison, for the monometallic catalyst MnCl_2_bpy in the presence of (EtO)_2_P(O)H (1:168), i_cat_/i_p_ = 3.6 (Table 1), and for the Ni(BF_4_)_2_bpy complex, i_cat_/i_p_ = 1.3 (Table 1). The cobalt catalyst regenerated even more slowly, which did not allow quantitative information to be obtained from CV. Preparative electrolysis data confirmed a greater efficiency of bimetallic catalytic systems for aromatic phosphorylation [34,35,36,37,38,39,40]. Thus, phosphorylation reactions of benzene with (EtO)_2_P(O)H were performed under electro-oxidation conditions using a 1% catalyst. The highest yields were obtained using MnCl_2_bpy/Ni(BF_4_)_2_bpy and MnCl_2_bpy/CoCl_2_bpy pairs as catalysts [34,35,36,37,38,39,40] (up to 71%) compared to monometallic catalytic systems Mn^II^(bpy), Ni^II^(bpy), and Co^II^(bpy) (~30%) [34,35,36,37,38,39,40].

### 2.2. Electroreduction Conditions

Formation of the previously published [42] aromatic phosphonation product under reduction conditions can be described by Scheme 4:

Hydrogen gas evolution was observed in this case. When comparing the catalytic activity of the studied catalysts under reduction conditions by cyclic voltammetry in the presence of DEP, it was found that, for all the complexes, catalytic growth was observed at one or another potential.

CoCl_2_bpy was reduced at 1.08 V. When adding increasing amounts of (EtO)_2_P(O)H a pre-wave irreversible peak appeared at 0.89 V (Appendix A). Reduction of the MnCl_2_bpy/CoCl_2_bpy complex mixture occurred at 0.92 V. When (EtO)_2_P(O)H was added, this peak shifted to the anode region by 30 mV (0.89 V) (Appendix A). In this case, it was also impossible to use the TOF calculation formula, since, as in oxidative conditions, the picture differed from the classical one, for which empirical formulas were derived.

Reduction of MnCl_2_bpy occurred at 0.90 V. With the addition of increasing amounts of (EtO)_2_P(O)H, the peak shifted by 20 mV to 0.88 V, and a catalytic growth was observed, where i_cat_/i_p_ (1:144) = 20 (Appendix A) and the TOF value was 344 s^−1^.

New peaks appeared on the CVA (cyclic voltammograms) of the cobalt complex in the presence of DEP and was attributable to the product of the chemical reaction of binding with DEP, cobalt phosphonate. These peaks are the single-electron (in oxidizing and reducing conditions) and does not grow with an increasing concentration of DEP. That is, regeneration of the cobalt catalyst was slow in CVA conditions. And for bimetallic systems, confident current growths were observed. That is, catalytic currents were at the redox potentials of metal phosphonates, which indicated a markedly higher catalyst regeneration rate in electrochemical reactions, so that it was possible to obtain quantitative information about k_obs_ or TOF.

Voltammetric data were previously described for silver catalysts operating in radical C–H phosphonation reactions, and no catalytic growth was observed under voltammetric conditions.

The assumed mechanisms of aromatic C–H bond phosphonation can be represented by the following scheme (Scheme 5):

The first stage was the interaction of the precursor of the catalyst active form with DEP with the formation of a new complex, which was then oxidized or reduced on the electrode with the elimination of the phosphonyl radical, which, in turn, reacted with the aromatic substrate. Two different intermediates after electron addition (or removing) were possible as the result of metal-centered or ligand-centered electron transfer. During the oxidation process, when the electron was removed, formation of a radical cation [M^2+^LS-H]^+•^ was possible when the oxidation degree of the metal was not changed, or a complex was formed with the changing of the metal oxidation degree [M^3+^LS-H]. Similarly, during the reduction process, electron transfer occurred with the participation of ligand orbitals with the formation of a radical anion [M^2+^LS-H]^−•^, or with the participation of the metal orbitals with a formation of [M^+^LS-H]. Hydrogen gas was released under reduction conditions, and H^+^ protons were produced under oxidizing conditions.

The radical mechanism of the aromatic C–H phosphonation under reduction and oxidation conditions was confirmed by the EPR method, using a cobalt catalyst as an example, when the (EtO)_2_(O)P^•^ radical was captured by a alpha-phenyl N-tertiary-butyl nitrone (PBN) spin trap [42]. The radical mechanism of the relative phosphorylation reaction in the presence of silver salts was confirmed also by the EPR method [20,44], which suggested that the reaction proceeded similarly when using other metal complexes or salts.

Thus, the combination of metal radical catalysis and electrochemically induced transformation provides certain advantages, both for the synthesis and for the establishment of regularities of the reactions. Electrochemical synthesis allows to obtain the products of CH/PH cross-coupling in a single stage under mild conditions in the absence of an excess of specially added oxidizing agents used in traditional synthesis methods in high yields (above 70%). Bimetallic catalysts demonstrate the best results, which was confirmed by kinetic studies.

## 3. Materials and Methods

### 3.1. Cyclic Voltammetry

Cyclic voltammograms were recorded with a BASi EpsilonE2P (USA) potentiostat. The device comprised a measuring unit, PC DellOptiplex 320 with the Epsilon-EC-USB-V200 software (BASi, West Lafayette, IN, USA). Tetrabutylammonium tetrafluoroborate (C_4_H_8_)_4_NBF_4_ was used as background electrolyte. A stationary disc glassy-carbon electrode with surface area of 6 mm^2^ was used as working electrode. The reference electrode was Ag/AgCl (0.01 M KCl), and the counter electrode was a platinum wire. The reference electrode was connected to the cell solution with a modified Luggin capillary filled with the supporting electrolyte solution of 0.1 M Bu_4_NBF_4_ in CH_3_CN. Thus, the reference electrode assembly had two compartments, each terminated with an ultrafine glass frit to separate the AgCl from the analyte. The auxiliary electrode was a platinum wire. The scan rate was 100 mV s^−1^. The measurements were performed in a temperature-controlled electrochemical cell (volume from 1 to 5 mL) in an inert gas atmosphere (N_2_). The solution was actively stirred with a magnetic stirrer between measurements, or prior to registration of a voltammetry wave, in the atmosphere of constant inflow of an inert gas, which was run first through a dehydrating system and then through a nickel-based purification system BI-GAS cleaner (manufactured by OOO Modern Laboratory Equipment, Novosibirsk) to remove trace quantities of oxygen.

### 3.2. Synthesis of the Metal Complexes

The complexes were synthesized by a known method [34,39,40]. The spectroscopic data for CoCl_2_bpy [55], Ni(BF_4_)_2_bpy [56], and MnCl_2_bpy [56] complexes matched that reported in the literature.

### 3.3. Data Processing of Voltammetric Measurements of Complexes

All voltammetric measurements were carried out at room temperature (≈23 °C). All measurements were performed at least three times. For the studied Co, Ni, and Mn complexes, CVs were recorded at various potential sweep rates (0.1–20 V s^−1^).

Determining the number of catalytic turns was carried out according to the Savéant formula [27]:
$$\frac{i_{cat}}{i_{dif}}=\frac{n}{0.4463}\sqrt{\frac{RT}{F}}\sqrt{\frac{k{\left[C\right]}^{2}}{v}}=0.72\sqrt{\frac{k{\left[C\right]}^{2}}{v}},$$
where k_obs_ = k [c]. TOF = k_obs_ when the rate ν did not depend on the concentration [C].

To establish the minimum scanning rate at which the increase in catalytic current did not depend on the concentration of the added *H*-phosphonate, a series of experiments were carried out for each metal complex in the presence of a phosphorylating agent. Solutions of metal complexes in acetonitrile were obtained in the presence of an excess amount of DEP. The CVs of the oxidation of the obtained solutions of the complexes were recorded at different scanning rates (0.1, 0.2, 0.4, 0.8, 1.6, 3.2, 6.4, 8.3, 10.0, 12.5, 14.28, and 20.0 V s^−1^). Based on the obtained results, a graph of the dependence of the increase in the catalytic current on the scanning rate was built (Appendix A). As can be seen from the graphs *I_cat_* vs. υ, the maximum scanning rates at which the current stopped increasing were: −10 V/s for MnCl_2_bpy, −12.5 V/s for Ni(BF_4_)_2_bpy, and −8.3 V/s for Ni(BF_4_)_2_bpy/MnCl_2_bpy.

## 4. Conclusions

Over the past decades, a C–H bond activation strategy has been successfully realized mainly using metals such as Pt, Pd, and Ag. From another side, the use of nonprecious, widespread metals such as Mn, Co, and Ni to activate the C–H bond seemed impossible until recently, despite the prevalence of these metals in the active center of various enzymes. As a rule, harsh reaction conditions were required to achieve their necessary reactivity. We proposed a new approach for the phosphonation of aromatic and heteroaromatic compounds, based on electro-oxidation or electroreduction, with the participation of widely distributed transitional metal complexes as catalysts. We estimated kinetic parameter (TOF) for all complexes used in these phosphorylation reactions. The highest value was obtained for the mixture of metal complexes Mn^II^bpy/Ni^II^bpy and Ni^II^bpy/Co^II^bpy (690 s^−1^ and 721 s^−1^, respectively) compared to individual complexes. This was also confirmed by preparative electrolysis data, where the product yielded up to 71% when bimetallic catalytic systems were used as a catalyst and ~30% when monometallic catalytic systems were used as a catalyst. In some cases (e.g., for Co^II^bpy and Mn^II^bpy/Co^II^bpy) it was impossible to calculate TOF since the new obtained wave on CV was not catalytic and was related to the formation of the new complex. So, voltammetry provides important information, not only about the redox properties of catalysts, but also their reactivity in electrocatalytic reactions of aromatic C–H phosphonation in the presence of a phosphorus precursor.

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
