# Peer review of "Evaluation of Transition Metal Catalysts in Electrochemically Induced Aromatic Phosphonation"

_molecules, 2019, doi:10.3390/molecules24091823_

Round 1
Reviewer 1 Report
The ms of Budnikova is on electrochemical aromatic phosphonation in the presence of new metal catalysts. As such, the ms is sound, however, it is a rather exhausting reading. It would be desirable to shorten the ms, and to transfer the less important details into the Supplementary Info. Typically, the parts around Figs 2-5 mean the “critical mass”. This Referee feels that a shorter discussion with Figs 2-5 in the Supplementary would be make this paper more attractive to read. Other shortcomings to be eliminated. - Scheme 2 is not clear-cut at all. The parts “A, B and C” would deserve 5 distinct schemes, for the 5 kinds of products. Part D is a quite different issue - this should be referred to in the text as a separate item. - Scheme 3 = Scheme 4. Why does it appear twice? In Scheme 5, pls highlight the tiny differences: “M3+LS-H” and “M+LSH” - Instsead of “phosphonation” would not it be better to use “phosphonylation”? - pls sum up the synthetic use of the method described to see the novelty. - minor issues: - p 3, line 89: “phosphor-centered” - p 4, line 142: “ or diarylphosphine oxides Ph2P(O)H” (?) - In conclusion, this ms may be accepted after the major revisions outlined above.
Author Response
Referee: 1
The ms of Budnikova is on electrochemical aromatic phosphonation in the presence of new metal catalysts. As such, the ms is sound, however, it is a rather exhausting reading. It would be desirable to shorten the ms, and to transfer the less important details into the Supplementary Info. Typically, the parts around Figs 2-5 mean the “critical mass”.
Q1: This Referee feels that a shorter discussion with Figs 2-5 in the Supplementary would be make this paper more attractive to read.
Response: Corrected, Figs. 2-6 were transferred to the Supplementary Information.
Q2: Other shortcomings to be eliminated. - Scheme 2 is not clear-cut at all. The parts “A, B and C” would deserve 5 distinct schemes, for the 5 kinds of products. Part D is a quite different issue - this should be referred to in the text as a separate item.
Response: Thank you for this comment, but Scheme 2 is generalized early published electrochemical reaction of C-H phosphonation. We clarified and described the details : “It was shown the C-H/P-H cross-coupling of dialkyl-H-phosphonate and different (hetero)aromatic molecules, such as benzenes bearing donor and acceptor substituents in the ring or coumarines (Scheme 2, A (oxidative conditions), B (reductive conditions)), azole derivatives (benzo-1,3-azoles, 3-methylindole, 4-methyl-2-acetylthiazole) (Scheme 2,C). Electrooxidative CH/PH functionalization of diphenylphosphine oxide with acetylenes in the presence of catalytic amounts of Ag+ yields to benzo[b]phosphole oxides (Scheme 2,D). The radical mechanism type was proposed for mentioned above electrocatalytic reactions, but the regularities and kinetics have not been established.”
Q3: - Scheme 3 = Scheme 4. Why does it appear twice?
Response: We corrected the scheme 4 (the misprint). There two different schemes for oxidation and reduction.
Q4: In Scheme 5, pls highlight the tiny differences: “M3+LS-H” and “M+LSH”
Response: Two different intermediates after electron addition (or removing) are possible as the result of metal-centered or ligand-centered electron transfer. During the oxidation process, when the electron is taken away, it is possible the formation of a radical-cation [M2+LS-H]+·, in which the oxidation degree of the metal isn’t changed, or the formation of a complex with the changing of the metal oxidation degree [M3+LS-H]. Similarly, during the reduction process, electron transfer occurs with the participation of a ligand or a phosphonate fragment orbitals with the formation of radical-anion [M2+LS-H]-·, or with the participation of the metal orbitals [M+LS-H]. We added this explanation.
Q5: - Instead of “phosphonation” would not it be better to use “phosphonylation”?
Response: Phosphonation (or phosphorylation) is a currently accepted designation of this reaction type.
Phosphonylation is misspelling of phosphorylation (https://en.wiktionary.org/wiki/phosphonylation).
Q6: - pls sum up the synthetic use of the method described to see the novelty.
Response:
Corrected. We concluded: “Electrochemical synthesis allows to obtain the products of CH/PH cross-coupling in a single stage under mild conditions in the absence of an excess of specially added oxidizing agents used in traditional synthesis methods, in high yields (above 70%), and bimetallic catalysts demonstrate the best results, what was confirmed by kinetic studies.”
Q7: - minor issues: - p 3, line 89: “phosphor-centered” - p 4, line 142: “ or diarylphosphine oxides Ph2P(O)H” (?) –
Response: Corrected.
Reviewer 2 Report
This paper proposed a new approach to phosphorylation of aromatic and heteroaromatic compounds based on electrooxidation or electroreduction with the participation of widely distributed transitional metal complexes as catalysts. Considering the novelty and importance, this manuscript is highly recommended for publication on Molecules after some minor revisions.
Some commence:
1 it will be more convincing to add the yields to the conclusion.
2 some related publications with dialkyl-H-phosphonates and diaryl phosphine oxides should be cited. For example, Pure Appl. Chem., 2019, 91(1), 33-41
Author Response
Referee: 2
This paper proposed a new approach to phosphorylation of aromatic and heteroaromatic compounds based on electrooxidation or electroreduction with the participation of widely distributed transitional metal complexes as catalysts. Considering the novelty and importance, this manuscript is highly recommended for publication on Molecules after some minor revisions.
Some commence:
Q1: it will be more convincing to add the yields to the conclusion.
Response: Corrected.
Q2: some related publications with dialkyl-H-phosphonates and diaryl phosphine oxides should be cited. For example, Pure Appl. Chem., 2019, 91(1), 33-41
Response: Corrected, this reference was included.
Round 2
Reviewer 1 Report
This Referee still beleives tht Sche is rather awkward in the present form and not copatible with the high standard of Molecule.
It is repeated:
" Scheme 2 is not clear-cut at all. The parts “A, B and C” would deserve 5 distinct schemes, for the 5 kinds of products. Part D is a quite different issue - this should be referred to in the text as a separate item."
Wh would it b difficultto replace this unfortunate illustrationwith precise ones. It is even not consistent re the dialkyl phosphite and diethylphosphite.
Emglish polishing i aso recommmended before final acception.
Author Response
Referee: 1
" Scheme 2 is not clear-cut at all. The parts “A, B and C” would deserve 5 distinct schemes, for the 5 kinds of products. Part D is a quite different issue - this should be referred to in the text as a separate item."
Why would it be difficult to replace this unfortunate illustration with precise ones. It is even not consistent the dialkyl phosphite and diethylphosphite.
Response: Corrected, Scheme 2 have been divided into three separated reactions according to the different catalytic conditions (catalyst, etc.) –A, B, C. The reaction D was cut out and is only mentioned in the text as reference 44.
Also, we rewrote some phrases and improved English.